# Establishment of a Faba Bean Banker Plant System with Predator *Orius strigicollis* for the Control of Thrips *Dendrothrips minowai* on Tea Plants under Laboratory Conditions

**DOI:** 10.3390/insects12050397

**Published:** 2021-04-29

**Authors:** Chang-Rong Zhang, Mei Liu, Fei-Xue Ban, Xiao-Li Shang, Shao-Lan Liu, Ting-Ting Mao, Xing-Yuan Zhang, Jun-Rui Zhi

**Affiliations:** 1Institute of Plant Protection, Guizhou Academy of Agricultural Sciences, Guiyang 550006, China; zhangchangrong2006@163.com (C.-R.Z.); liumei721122@163.com (M.L.); lan605605@163.com (S.-L.L.); maotingting2922@163.com (T.-T.M.); stirm_zhangxy@163.com (X.-Y.Z.); 2Institute of Entomology, Guizhou Provincial Key Laboratory for Agricultural Pest Management of the Mountainous Region, Guizhou University, Guiyang 550025, China; zhijunrui@126.com; 3Key Laboratory of Biology and Medical Engineering, School of Biology and Engineering, Guizhou Medical University, Guiyang 550025, China

**Keywords:** alternative prey, biological control, *Aphis fabae*, control efficacy, *Camellia sinensis*

## Abstract

**Simple Summary:**

The banker plant system may provide an effective and economical method for long-term suppression of insect pests. We developed a non-crop banker plant system aiming to improve the control of stick tea thrips *Dendrothrips minowai* in tea plantations. In this system, we used the polyphagous predator *Orius strigicollis* (Poppius) as the biocontrol agent, the black bean aphid *Aphis fabae* (Scopoli) as an alternative food, and the faba bean *Vicia faba* L. as the banker plant to support the predator in controlling pest thrips. Laboratory tests revealed that the control efficacy of the banker plant system was higher than that of directly releasing *O. strigicollis*. These results indicate that this banker plant system may be used in the field to provide a more effective and economical way to control pest thrips in tea plantations compared with the direct release of *O. strigicollis*.

**Abstract:**

The stick tea thrip *Dendrothrips minowai* (Priesner) (Thysanoptera: Thripidae) is a destructive pest in tea plantations in south and southwest China. To control this pest, a non-crop banker plant system was developed using a polyphagous predator *Orius strigicollis* (Poppius) (Heteroptera: Anthocoridae) with the black bean aphid *Aphis fabae* (Scopoli) (Hemiptera: Aphididae) as an alternative prey and the faba bean *Vicia faba* as the banker plant to support the predator in targeting the pest. The fitness of *A. fabae* on tea plants and faba bean was evaluated to determine its host specificity. Moreover, the control efficacy of the banker plant system on *D. minowai* on tea plants was tested in the laboratory and compared with that of direct release of *O. strigicollis*. The experiments showed that faba bean was an excellent non-crop host for *A. fabae* because, while the aphid population increased quickly on faba bean, it could only survive for up to 9 days on tea plants. Compared with direct release of *O. strigicollis*, lower densities of pest were observed when introducing the banker plant system. Our results indicate that this banker plant system has the potential to be implemented in the field to improve the control of the pest thrips.

## 1. Introduction

Tea is a traditional Chinese beverage, and the cultivated area of tea plant *Camellia sinensis* (L.) O. Kuntze (Theaceae) in China reached 45.75 ha in 2020. Tea plants are prone to infestations of pests and diseases, which lead to losses in harvest yield and quality. The stick tea thrip *Dendrothrips minowai* Priesner (Thysanoptera: Thripidae) is one of the main pests that causes serious damage in tea plantations. This species is mainly distributed in south and southwest China, especially Guizhou Province [1,2]. Adults and nymphs of *D. minowai* damage the tender leaves using rasping–sucking mouth parts, causing leaf abscission, wrinkling, and growth retardation, which seriously impact tea quality and yield [3]. According to previous studies, control methods for *D. minowai* primarily include timely weeding, pruning, and application of pesticides (e.g., azadirachtin, chlorfenapyr, and bifenthrin) [3,4]. Due to pesticide resistance and residue problems, increasing attention has been focused on integrated pest management based on biological control. Recently, biological control of *D. minowai* by natural enemies, such as the predatory mite *Amblyseius cucumeris* (Oudemans) (Acari: Phytoseiidae) and the ladybeetle *Harmonia axyridis* (Pallas) (Coleoptera: Coccinellidae), has been reported [4]. However, many other natural enemies and their potential applications have not been studied.

Flower bugs of the *Orius* genus (Hemiptera: Anthocoridae) are among the most effective predators of thrips, whiteflies, mites, and other small pests. To date, many *Orius* species have been successfully released as predators to control pests, for example, *O. niger* (Wolff) and *O. laevigatus* (Fieber) against *Frankliniella occidentalis* (Pergande) in Turkey [5], and *O. minutus* (L.) and *O. niger* (Wolf.) against *Odontothrips loti* (Hal.) in Romania [6]. In addition, flower bugs such as *O. laevigatus*, *O. insidiosus*, and *O. majusculus* have been commercialized in the Americas and European countries [7,8,9]. Among *Orius* spp., *O. strigicollis* (Poppius), which was also known as *O. similis* (Zheng) [10], has been found to effectively control various pests, especially thrips and aphids [11,12,13]. It is also a potential biological control agent of *D. minowai*.

Although many studies have shown that releasing natural enemies can effectively control pests (e.g., [14,15,16])**,** the requirement for large numbers of natural enemies and high cost have limited application. In addition, poor establishment and persistence of natural enemies significantly influence the sustainability of control efficacy. It is imperative to find methods to overcome these problems. One possible solution is the “banker plant system” (also called the open-rearing system), which is a natural enemy rearing and release system typically consisting of banker plant, alternative food, and beneficial insects (predators or parasitoids) [17]. Banker plants grow together with the primary crop and provide alternative food such as pollen or non-pest herbivores to promote the survival and reproduction of natural enemies. Therefore, natural enemies can be established on the banker plant and then target specific pests without the need to repeatedly release these natural enemies. As the banker plant system can offset the drawbacks of augmentative and conservation biological control, it has been applied in pest control in greenhouses or fields in several countries [17,18,19,20]. Banker plant systems with flower bugs as beneficial insects have also been developed, for example, ornamental pepper (*Capsicum annuum*) as a banker plant providing pollen as alternative food for support of *O. insidiosus* for the control of *F. occidentalis* in ornamental grass [21,22], and *Calendula officinalis* as a banker plant supplying extrafloral nectaries as an alternative food for the support of *O. sauteri* for the control of *F. occidentalis* in tomato [23]. However, use of the flower bug–banker plant system for the control of tea pests is less common.

Faba bean (*Vicia faba* L.) is a host plant of *Aphis fabae*, which is a suitable aphid prey for flower bugs [24,25]. Although *A. fabae* is polyphagous, it prefers legumes, and we speculate that it may not infest tea plants. Furthermore, faba bean can provide some benefits to flower bugs, such as nectar, pollen, shelter, and egg-laying sites [26]. Therefore, faba bean was considered as a potential banker plant to maintain *O. strigicollis* through providing *A. fabae* as an alternative food.

The goal of this study was to develop a novel *O. strigicollis*-based banker plant system for the control of *D. minowai*. Specifically, we evaluated (a) the fitness of *A. fabae* on tea plants to determine if *A. fabae* is a non-pest for tea plants and (b) the use of the banker plant system consisting of *O. strigicollis–*faba bean–*A. fabae* for the control of *D. minowai* on tea plants. In addition, we compared the efficacy of this system with that of the direct *O. strigicollis* release.

## 2. Materials and Methods

### 2.1. Insects and Plants

Colonies of *O. strigicollis* were established from specimens collected from corn fields at Guizhou Academy of Agricultural Sciences, Guizhou, China. The species identity was determined using the method of Zhang et al. [27]. *O. strigicollis* were reared with *Frankliniella occidentalis* adults and *Sitotroga cerealella* eggs as prey for 4–6 generations. The *D. minowai* were collected from a tea plantation of the Guizhou Academy of Agricultural Sciences and reared on tea plants for 4–6 generations. *A. fabae* was provided by the Institute of Plant Protection, Guizhou Academy of Agricultural Sciences, and reared on 2-week-old faba bean seedlings. The apterous aphids reproduced by cyclical parthenogenesis. The annual seedlings of tea (Qiancha No. 1) were collected from a tea plantation of Meitan County, Guizhou Province. Then, 2–3 tea seedlings were planted in a small pot (length × width × height: 10 cm × 8.5 cm × 10 cm). Faba beans used as banker plants in the experiment were prepared according to the following procedure. First, faba bean seeds were soaked in water for 5–6 h, and the excess water was then removed. Next, all faba bean seeds were placed in a bamboo basket and moisturized by covering with a wet towel. After 1/3 of the faba bean seeds had sprouted, they were placed in a climate chamber for vernalization for 14 d (4 ± 1 °C and RH 70 ± 5% in darkness). Faba bean seeds were planted in small pots (three seeds per pot) after vernalization and used for experimentation when the seedlings reached a height of 10–15 cm. All experiments were conducted at 25 ± 1 °C, 70 ± 5% relative humidity, and 16:8 h of light (L):dark (D) in an air-conditioned room.

### 2.2. Evaluating A. fabae Fitness on Faba Bean and Tea

Amounts of 5, 10, and 15 *A. fabae* adults were separately introduced into three pots of annual tea seedlings, and 5 *A. fabae* adults were introduced into a pot of faba bean seedlings. Then, the tea plants and faba bean seedlings infested with *A. fabae* were maintained in separate cages (30 cm × 30 cm × 30 cm). *A. fabae* numbers were counted daily until all had died. Each treatment was replicated three times.

### 2.3. Evaluating the Initial A. fabae Population to Maintain O. strigicollis

Five female *A. fabae* adults (3 days post-emergence) were released into each of the four 120-mesh insect-proof cages with one pot of faba bean (approx. 10–15 cm tall). Then, five female *O. strigicollis* adults (3 days old) were introduced into each of the three cages with *A. fabae* after 24, 48, and 72 h, respectively. Cages without *O. strigicollis* were used as control. The numbers of *A. fabae* were counted daily at 16:00 for 10 days. Each treatment was replicated three times.

### 2.4. Evaluating Control Efficacy of Banker Plants on D. minowai

Four treatments were conducted: (1) In the *O. strigicollis–V. faba–A. fabae* banker plant system, one pot of *V. faba* and six pots of tea seedlings were placed in a muslin cage (length × width × height of 60 cm × 45 cm × 45 cm; the cage size was the same for treatments 1–4). Five *A. fabae* were introduced into the cage for 72 h before releasing female *O. strigicollis* adults, and 50 *D. minowai* were then introduced into the cage after 10 days. (2) For direct release of *O. strigicollis*, five female *O. strigicollis* adults were released into the muslin cage with six pots of tea seedlings, and 50 *D. minowai* were then introduced into the cage after 10 days. (3) For the banker plant system without *O. strigicollis*, five *A. fabae* were released into a cage with one pot of *V. faba* and six pots of tea seedlings, and 50 *D. minowai* were then introduced into the cage after 10 days. (4) For thrips only, 50 *D. minowai* were introduced into a cage with six pots of tea seedlings. All mated female *O. strigicollis* adults were used in the experiment at 3 days post-emergence. *D. minowai* numbers were counted weekly for 4 weeks. Each treatment was replicated four times.

### 2.5. Statistical Analyses

Since the data were not distributed normally, significant differences in the number of *D**. minowai* were analyzed using non-parametric Kruskal–Wallis tests followed by Dunn–Bonferroni post hoc pairwise comparisons (SPSS 25.0; IBM Company, Armonk, NY, USA). Figures were drawn using GraphPad Prism 8 (GraphPad Software Inc., San Diego, CA, USA).

## 3. Results

### 3.1. Host Specificity of A. fabae

When *A. fabae* adults were reared on tea seedlings, they all died after 4, 6, and 9 days when the initial numbers of adults were 5, 10 and 15, respectively; by contrast, the number of *A. fabae* reared on faba bean increased steadily after the initial release of five adults and reached about 16 times that of day 1 (Figure 1). These results indicate that *A. fabae* have poor survival on tea plants.

### 3.2. Initial A. fabae Population to Maintain O. strigicollis

Five *A. fabae* were released onto faba bean seedlings and allowed to feed and reproduce freely for 24, 48, and 72 h to obtain different sizes of initial population. The steady increase in aphid number in the treatment without *O. strigicollis* indicates that these plants are a suitable host for the aphid (Figure 2). In the two treatments where the aphids were left to feed and reproduce for 24 and 48 h, the mean numbers of aphids increased to 18 (3.6 times higher than the original) and 32 (6.4 times higher than the original), respectively. However, the numbers of aphids deceased steadily and down to zero at 4 and 6 days after release of the *Orius* adults, respectively (Figure 2). In the treatment where the aphids were left to feed and reproduce for 72 h, the mean number of aphids increased to 46 (9.2 times higher than the original), and number of aphids declined slowly after the release of *Orius* adults and remained at around 20 from day 7 onwards (Figure 2).

### 3.3. Efficacy of Banker Plant System for the Control of D. minowai

In the *O**. strigicollis–V**. faba–A**. fabae* banker plant system, *A. fabae* and *O. strigicollis* were pre-released on faba bean seedlings for 10 days prior to the release of 50 *D. minowai* onto the tea seedlings. The results of Kruskal–Wallis testing revealed that there was a significant difference in the number of *D. minowai* among treatments (21 d: χ^2^ = 8.609, df = 3, *p* = 0.035; 28 d: χ^2^ = 8.809, df = 3, *p* = 0.032; 35 d: χ^2^ = 8.597, df = 3, *p* = 0.035; 42 d: χ^2^ = 9.486, df = 3, *p* = 0.023; 49 d: χ^2^ = 9.336, df = 3, *p* = 0.025) (Figure 3). However, the Dunn–Bonferroni post hoc test indicated the difference between each of the two treatments was not statistically significant from 21 to 35 d. Compared with the banker plant system without *O. strigicollis* and thrips only treatments, the number of *D. minowai* in the treatment of the banker plant system was significantly lower at 42 d (Figure 3). Moreover, we found that in the treatment of banker plant system, the number of *D. minowai* approached 0 throughout the experiment, which was lower than that of the other three treatments (Figure 3).

## 4. Discussion

Although the banker plant system has been applied to control pests such as whiteflies, aphids, and thrips, much work is still needed to screen different banker plant components to target different crops and pests. Here, an *O. strigicollis*-based banker plant system was established for its potential application in the control of the tea pest thrips *D. minowai*. In the established system, faba bean is a non-crop host plant for *A. fabae*, which serves as alternative prey to support the beneficial insect *O. strigicollis*.

In a successful banker plant system, alternative preys that are specific to the banker plant are preferable in order to eliminate the risk that the crop becomes infested by the herbivore. *A. fabae* could only survive for a few days on tea plants but reproduced quickly on faba bean (Figure 1), indicating that *A. fabae* is both a non-pest alternative herbivore for maintaining the predator *O. strigicollis* and not a pest for tea plants. In addition, there are three major advantages to using faba bean as banker plants. First, faba bean is easy to grow in the field and greenhouse and can be heavily fed on by *A. fabae*, which can provide a long-term food source for *O. strigicollis*. Second, faba bean has a high ability to fix nitrogen (N) from the atmosphere and can therefore provide a source of N for the crop [28]. Third, faba bean may attract other natural enemies, such as *Aphidoletes aphidimyza* (Diptera: Cecidomyiidae) and *Lysiphlebus fabarum* (Hymenoptera: Braconidae: Aphidiinae), through provision of *A. fabae* as a food [29,30]. It is possible that faba bean could serve as a banker plant to support multiple natural enemies to target different pests. Moreover, according to our previous laboratory study, *O. strigicollis* egg production is higher on faba bean than that on *Rosa chinensis*, *Narcissus jonquilla*, *Capsicum annuum*, and *Cucumis sativus* [31], indicating that faba bean can simultaneously satisfy the reproduction of natural enemy and alternative prey. Moreover, Hansen et al. (1983) used faba bean as a banker plant, providing *Megoura viciae* (Hemiptera: Aphididae) as alternative food to support *A**.** aphidimyza* for the control of *Myzus persicae* in sweet pepper [32]. Faba bean can also supply *Megoura japonica* (Hemiptera: Aphididae) as alternative food for the support of *Coccinella septempunctata* (Coleoptera: Coccinellidae) in controlling *Aphis gossypii* (Hemiptera: Aphididae) (unpublished).

According to a previous study, a long pre-plant release period can significantly increase the abundance of predator *Macrolophus pygmaeus* (Hemiptera: Miridae) and its efficiency in controlling *Bemisia tabaci* (Hemiptera: Aleyrodidae) and *Tuta absoluta* (Lepidoptera: Gelechiidae) on tomato [33]. In this study, an interval of 10 days that was similar to the pre-plant release period was used. During the 10 days, *O. strigicollis* only fed on the *A. fabae* on faba bean, which may have increased the adaptation and initial population of the predator, thus allowing it to successfully establish. Moreover, one of the key factors influencing the establishment of a natural enemy is the amount of prey, so a laboratory experiment was performed to estimate the size of the initial *A. fabae* population required to maintain *O. strigicollis* under conditions without the pest *D. minowai*. The results showed that the number of *A. fabae* reached 46 after 72 h of reproduction, which could maintain the initial population of five *O. strigicollis* for at least 10 days and allow the number to be maintained at a relatively high level (Figure 2).

Although there was no statistically significant difference, the number of *D. minowai* in the *O. strigicollis* banker plant system was found to be lower than that when *O. strigicollis* were directly released throughout the experiment (Figure 3). These results are similar to those reported by Wong and Frank (2011), who established an *O. insidiosus*–black pearl pepper banker plant system and found that its control efficacy was higher than that of direct *O. insidiosus* release [22]. In this experiment, two treatments without *O. strigicollis* (i.e., the banker plant system without *O. strigicollis* and thrips only) were also evaluated to analyze interactions between plants (faba bean and tea) and alternative prey and pests. These treatments were also used as controls for the two treatments with *O. strigicollis* (i.e., the *O. strigicollis*–*V. faba*–*A. fabae* banker plant system and direct release of *O. strigicollis*). The possible reason for the result is that *O. strigicollis* may became pre-established due to the banker plant system providing alternative prey and shelter. However, direct release resulted in reduced survival of *O. strigicollis*, likely due to a lack of food.

The present study was conducted under controlled conditions in a laboratory, which may limit the applicability of the results to the field. However, the results obtained here indicate that this established *O. strigicollis* banker plant system has great potential for field application. Atakan (2010) showed that *Orius* will move from overwintering sites, such as weeds to faba beans, when the temperature increases during the period of from March to April [26]. According to the previous investigation, *O. strigicollis* was found on faba bean in March, and tea pests, such as thrips, leafhoppers, and whiteflies, began to appear in tea plantations in April. Therefore, interplanting faba beans in tea plantation in February may attract *O. strigicollis* prior to target pest occurrence. The alternative prey *A. fabae* may play a positive role in the establishment of *O. strigicollis*. While the direct release method requires the purchase and release of a large number of *O. strigicollis*, which is expensive and time-consuming, there is no commercial *O. strigicollis* available. The production of predators from banker plants could reduce the high cost of releasing natural enemies. The effectiveness of the *O. strigicollis* banker plant system in commercial tea plantations needs further study. Additionally, the efficacy of banker plant systems is also affected by factors such as the spatial arrangement, season, release rate, and spreading ability of the natural enemy, all of which require further study.

## 5. Conclusions

The results of this study indicate that it may be possible to use the banker plant system to control *D. minowai* as well as other tea pests in the field. This strategy may be attractive to growers as it provides a more effective and economical way to control pests than direct the release of *O. strigicollis*.

## Figures and Tables

**Figure 1 insects-12-00397-f001:**
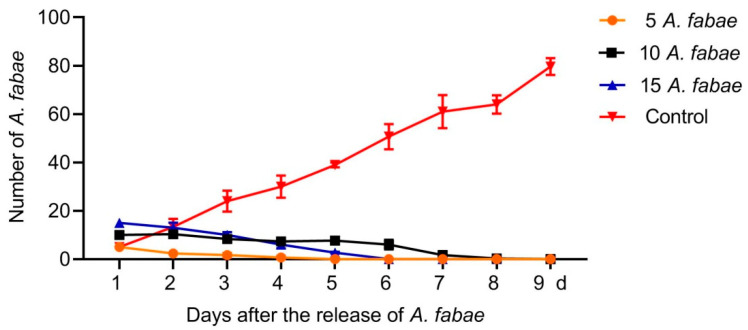
Numbers of *Aphis fabae* on tea or faba been plants on different days after the initial release of aphid adults. Five, ten, and fifteen *A. fabae* adults were released into each of the three insect cages containing one pot of tea seedlings. Five *A. fabae* adults were reared on three *Vicia faba* seedlings as a control. Numbers of *A. fabae* were counted daily until all *A. fabae* on tea seedlings died. Data are presented as mean ± SEM.

**Figure 2 insects-12-00397-f002:**
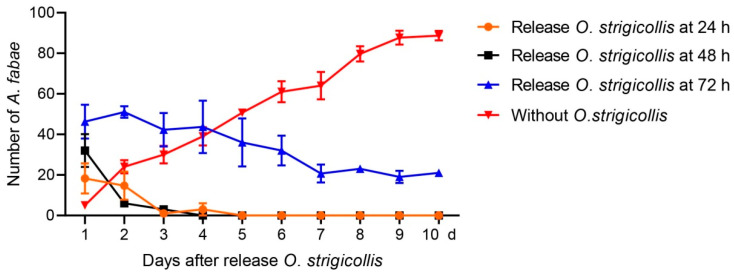
Number of surviving *Aphis fabae* in different treatments after the release of *Orius strigicollis* adults. Five *Orius* adults were introduced when the aphids had been feeding and reproducing on faba bean seedlings for 24, 48, and 72 h, and the treatment of aphids without release of *Orius* was used as control. Numbers of *A. fabae* were counted daily for 10 days.

**Figure 3 insects-12-00397-f003:**
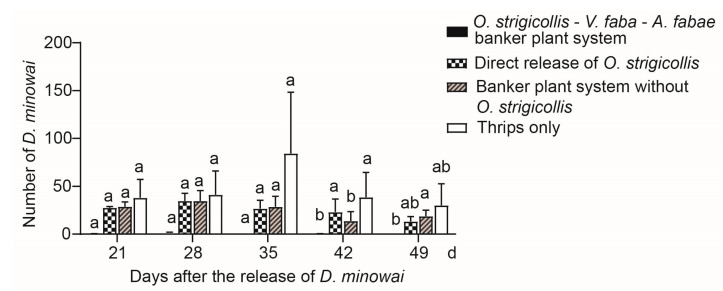
Number of *Dendrothrips*
*minowai* on tea seedlings in the four treatments: (1) *O**. strigicollis–V. faba–A. fabae* banker plant system, (2) direct release of *O. strigicollis*, (3) banker plant system without *O. strigicollis*, and (4) thrips only. Numbers of *D. minowai* were counted weekly for 4 weeks after releasing. Data are presented as mean ± SEM. Different letters indicate significant differences among the four treatments separately in each of the dates (*p* < 0.05) calculated using Kruskal–Wallis tests followed by Dunn–Bonferroni tests.

## Data Availability

The data presented in this study are available in article.

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
