# Peer review of "Establishment of a Faba Bean Banker Plant System with Predator Orius strigicollis for the Control of Thrips Dendrothrips minowai on Tea Plants under Laboratory Conditions"

_insects, 2021, doi:10.3390/insects12050397_

Round 1
Reviewer 1 Report
Please see my comments and suggestions in the attachment.

Reviewer 2 Report
In this study, Zhang et al., developed a non-crop banker plant system to improve the D. minowai controlling. In the system, they used the predator O. strigicollis as the biocontrol agent, the black bean aphid as an alternative food. The efficacy of this banker system was higher than directly releasing predator in the lab condition, which suggest it may be used to provide a more effective and economical way to control D. minowai in the field.
I have some comments/questions to improve this study, please see below:
- This study was conducted in the laboratory, if it’s possible, it would be better to test in green house and field.
- Line 132, why this time points were selected?
- Line 134, “16:00 h” change to “16:00”
- Line 147, “3 days” change to “three days”
- Line 165-166, please mention control group was reared on V. faba in both text and figure legend
- Figure 2, why the initial number of A. fabae was different among groups?
- Section 3.3, please add more description of Table 1, such as why there is a population increasing in population in 35 days?
- I think some of the discussion parts should put in result section and a deeper discussion is needed.
Reviewer 3 Report
Overall, I think this is a nice clean little study to document the use of banker plant systems. Given some unequal variance, I think the non-parametric approach seems appropriate. What I'd like to see is a little deeper dive into interpreting the results in the discussion. I think the discussion is pretty general/superficial and doesn't really work through the results and fully relate to recent works on banker plant systems, and aside from the thrips aspects, what we actually learn here to help advance the use of banker plant systems. There appears to be some timing elements, and also just having the banker plants there appears to provide some decrease in populations. There are also a few minor style things, so might need to go over the manuscript again to catch all. I highlighted a few.
One result that seems to need further explanation and description of methods is the results presented in Table 1. Some of the figures also require some work to know what is really being displayed so a bit more description there.

Round 2
Reviewer 2 Report
The revised version was greatly improved. I suggest accept in present form.
Author Response
We appreciate your positive evaluation on our study and constructive comments.